# OpenReview forum: "The Social Welfare Function Leaderboard: When LLM Agents Allocate Social Welfare"
_ICLR.cc/2026/Conference — ICLR 2026 Conference Withdrawn Submission_

### Official Review · Reviewer_4jz5 · 2025-10-28

**Soundness:** 2
**Presentation:** 2
**Contribution:** 2
**Rating:** 4
**Confidence:** 3

**Summary:**

The paper introduces a benchmark where an allocator LLM assigns tasks to a pool of smaller models over many rounds. They measure "efficiency" by ROI from the productivity of those models; fairness is based on the Gini coefficient; and the main score is their product. The authors find that existing chat leaderboards poorly predict allocation behavior. Findings also include that shorter outputs tilt allocators toward efficiency at the expense of fairness, and social-influence prompts can shift allocations toward fairness with efficiency loss.

**Strengths:**

1. The task is clearly defined and easily verified and reproduced.
2. The efficiency-fairness trade-off is particularly valuable to track, showing how LLMs (or AI systems generally) manage complex social values.
3. The study identifies practical levers that change behavior, such as shorter reasoning and social framing.

**Weaknesses:**

1. I am confused why this is a leaderboard. Are models meant to get higher on this? Is that really a good thing? Why can't a model be justified in prioritizing not fairness, efficiency, or the product of the two, but something else? I'm excited about benchmarks that tackle complex social topics like social welfare allocation, but I worry this is not the right framing. It may be better framed as a "Tracker". This also means I take issue with the terminology of "welfare allocation skill" since the models could be skilled but just not normatively aligned with the authors.
2. It seems there are many idiosyncrasies of prompting that could be shaping these results (e.g., "only the three most recent turns of history are retained as input"). The contribution would be stronger if the ranking of models were shown to be robust to such details. For example, the authors could build alternative systems that get at the same fairness vs. efficiency trade-off and identify whether results are robust to that. I am not convinced by the paper that we are really finding out durable, interesting features of different models.
3. I don't see why there is an "Algorithm" in this paper. It doesn't seem to contribute anything beyond restating the LLM decisions and scoring criteria. There is no sophisticated algorithm at play here, as far as I can tell.
4. Some of the explanation in the paper is unclear, such as "When making assignment decisions, each recipient agent is initialized
with its public rating from the LLM Arena." What does this mean? Why is this benchmark being influenced by the results of a different leaderboard?

Typos/etc:

- Language is often unclear in the paper, such as the meaning of, "Think less, then more utilitarian." I think the paper needs substantial copyediting.
- "Tempatation" -> "Temptation"
- "hihger" -> "higher"
- "totol" -> "total"
- "overal" -> "overall"
- "Clause‑Opus‑4.1" → “Claude‑Opus‑4.1"

**Questions:**

1. Is there non-determinism in the behavior of the "smaller LLM with heterogeneous capabilities"? How is this accounted for in the leaderboard ratings, since it could mean a lot of randomness goes into model placement?
2. Are the various findings in the paper based on significance tests? I am unclear whether results (other than Table 5) are being reported only with statistical significance.
3. What was meant by the "persuasion strategies" and other perturbations of input for the leaderboard? There seem to be important details missing from the paper.cross task flows and seeds.

---

### Official Review · Reviewer_G13H · 2025-10-29

**Soundness:** 2
**Presentation:** 3
**Contribution:** 2
**Rating:** 4
**Confidence:** 4

**Summary:**

This paper creates a task where LLMs act as resource allocators over a small society. The model is instructed to balance fairness and efficiency, which are scored with Gini coefficient and ROI respectively. The two metrics are multiplied together to produce an overall score. 20 leading LLMs are evaluated on this task, and the results show low correlation to performance on other benchmarks.

**Strengths:**

This paper introduces a new game-theoretic environment for assessing the behaviour of LLMs. The presentation of the work is generally clear, and raises an important question about the behaviour of LLMs in decision-making roles. The task is well chosen to test out the inherent tension between distributional fairness and overall efficiency.

**Weaknesses:**

This paper is oddly positioned in that it is considering an area where there an long-standing disagreements about what an optimal balance between efficiency and equality should look like, but somewhat arbitrarily establishes a one-dimensional benchmark score.

Specifically:

- This metric has odd marginal behaviours. Gini coefficients are not uniform, in that in some conditions reallocating a single task from an agent with more tasks to one with less has different marginal impacts than in other conditions. Mutiplying the two metrics together also ensures that the optimum will be a balance between the two metrics under most realistic conditions. (Relatedly, Figure 3 would be better with contours showing equal overall scores.)

- There isn't a clear normative argument that higher scores on this benchmark are preferable. There exists a theoretical maximum possible score which is determined by the relative abilities of the model population (and it isnt' 1.0), but this is largely arbitrary and not the result of a principled stance. Since higher scores are not necessarily better, I question whether this is actually a benchmark.

- The implied focus on a specific way of weighting the two sub-metrics isn't necessary for the task to be useful. I would rather have seen how well the LLM can execute on a given target Gini and efficiency, more similar to the prompting tests at the end. This alternative setup would allow humans to retain the moral agency of choosing how they prefer to balance these objectives rather than delegating that responsibility to the LLM.

There are experimental choices which don't seem helpful to the clarity of the paper:

- What does the benchmark gain by using actual models and real tasks for the population rather than just having hidden probabilities of success? Clearly a lot of effort was put into scoring these tasks that doesn't add much informational value to the results.

Some of the results are not well-supported by the evidence:

- In line 349, a claim is made that top models balance their allocation between the two metrics. This is circular, since not doing so would mean they were not top models (as a result of the multiplicative headline score).

- Paragraph 362-377 is not conclusive. To show this would require an experiment showing that the same result occurs when using different ELO scores for initial seeding.

- Line 422 implies an empirical validation of a fairness/efficiency tradeoff that exists by construction and needs no empirical validation.

The related work is too broad and doesn't cover the actual related work. Paragraph 444-456 is way too broad and  lacks scientific evidence for claims such as "richer forms of intelligence". 458-469 is also too broad. There are existing works that evaluate the behaviour of LLMs in game theoretic settings which would be the appropriate comparison here such as https://papers.ssrn.com/sol3/papers.cfm?abstract_id=4493398.

There are small details which are missing:
- Lines 205-212 need citations.
- How many repetitions were run? What are the error bars on these results?
- It would be worth checking that there are not innate biases in choosing between the assigned LLM names. (Is the "AAA" a more likely generation than "BBB" even without the context of the game?)
- There are spelling errors in the prompts (e.g. lines 1042-43)
- The ROI scores have high/low ratings but the task counts do not. Couldn't this bias the results?

**Questions:**

1. Do these results depend on the language in which the model is prompted? For example, does prompting in Chinese lead to different balances?

2. Why use multiplicative scoring? How sensitive are your results to the method of aggregation? (e.g. what if you had used min(fairness, efficiency)?)

3. Why is this metric the right one for testing social welfare, rather than a more pluralistic approach?

---

### Official Review · Reviewer_maQk · 2025-10-31

**Soundness:** 1
**Presentation:** 2
**Contribution:** 2
**Rating:** 2
**Confidence:** 4

**Summary:**

This paper introduces the Social Welfare Function benchmark, a simulation framework for evaluating how LMs allocate limited resources among agents with varying abilities, balancing efficiency and fairness. In the benchmark, an LLM acts as a allocator distributing tasks to other LLMs, where efficiency is measured by return on investment and fairness by the Gini coefficient and their product defines the SWF Score. Testing a variety of SOTA models, the authors find that performance on Chatbot Arena poorly predicts welfare allocation skill (measured by SWF) and that most LLMs exhibit a strong utilitarian bias, favoring efficiency over equality. Other experiments show that the results are sensitive to prompting. These results highlight that current LLMs lack stable ethical judgment in distributive contexts, underscoring the need for specialized benchmarks and alignment methods for AI systems acting in governance or societal decision-making roles.

**Strengths:**

- As far as I can tell, this is the first paper studying LLMs as allocators on this specific type of task.

- The experiment setup is not toy, several existing LM benchmarks are used to source questions and real models are used to generate scores from the recipient group.

- The experiments span a wide variety of closed and open source models.

- The metrics used are simple and easily interpretable (Gini, ROI).

**Weaknesses:**

- The paper contains no statistical tests to understand whether the claims are statistically significant, instead opting for presenting large amounts of data in tables and using subsets of the data to draw conclusions (e.g. pointing out “DeepSeek-V3-0324, ranked 25th on Arena, rises to 1st place on SWF with a score of 30.13”). Providing correlations between values and performing statistical tests to validate claims should be a baseline for this type of work and would strengthen the paper. This is my biggest issue with the current version of the paper.

- Line 346: “Top Arena models prioritize efficiency but lead to severe inequality”
This point is poorly supported. The top 4 arena models (ranked 1 or 2) all achieve below 52 on efficiency which is significantly lower than the most efficient models.

- The paper presents a lot of results in large tables but no summary statistics to draw conclusions from the data. For example, computing the correlation between Arena Score and SWF Score would allow the reader to much better understand the data.

- The random strategy performs better in SWF than most models ,potentially indicating that this task isn’t well suited to LMs.

- The choice of aggregating fairness and efficiency using a product seems somewhat arbitrary and the results seem like they would be highly dependent on the choice of aggregation. It would help to explain how the SWF metric connects to existing measures of social welfare or otherwise provide justification for the metric.

- The paper includes no error bars or measures of variance on the aggregate values. For example, the average row in Table 2. This makes it difficult to gauge the statistical significance of the average values, which is especially important given that there is high variance across different models.

- The paper investigates the fairness-efficiency relationship on just a single type of task, which limits the generality of several of the claims about LM behavior in general. It’s difficult to conclude how much of the results are dependent on the specific evaluation setup and details of the prompts or simulation environment.

**Questions:**

- Line 395: “LLMs naturally produce long chains of reasoning z before selecting a recipient, a behavior reminis-cent of human decision-making where extended deliberation may increase hesitation under decision fatigue“
Why is this reminiscent of decision fatigue? Can you support the idea that the model is hesitating?

- The random strategy performs better in SWF than most models, does this indicate that the models are generally bad at this routing task? If so, do you know why the models perform so poorly? This worse than random performance makes me worried that the domain is too difficult for the models and not suitable for properly evaluating the fairness/efficiency tradeoff.

- The results related to prompt sensitivity are interesting but also raise the concern that the results might not be generalizable across different settings. Do you think the results might change significantly if other parts of the problem setup were altered? Or if other parts of the prompts are changed?

- Only the last 3 rounds of dialogue are kept in the experiments. This seems like it could significantly affect results since more rounds of dialogue would allow the models to better understand the performance of the recipients on different tasks. Did you explore this at all?

---

### Official Review · Reviewer_fw5Z · 2025-11-05

**Soundness:** 2
**Presentation:** 3
**Contribution:** 2
**Rating:** 4
**Confidence:** 3

**Summary:**

This paper introduces the Social Welfare Function (SWF) Benchmark, a dynamic simulation environment designed to evaluate how Large Language Models (LLMs) navigate resource allocation decisions when acting as sovereign allocators. In this framework, an LLM distributes tasks sequentially to a heterogeneous community of recipient agents (smaller LLMs with varying capabilities), creating a persistent trade-off between maximizing collective efficiency (ROI) and ensuring distributive fairness (Gini coefficient).

The authors evaluate 20 state-of-the-art LLMs and and find that general conversational ability, as measured by popular leaderboards like Arena, is a poor predictor of allocation competence; most LLMs exhibit a strong default utilitarian orientation, prioritizing group productivity at the expense of severe inequality; and allocation strategies are highly vulnerable to external influences, being easily perturbed by output-length constraints and social-influence framing such as threats and temptations.

**Strengths:**

This paper makes a thoughtful attempt at creating an environment that simulates the mechanics of resource distribution and capturing a conflict between efficiency and fairness that could be representative of decision making tasks like hiring. The long-horizon nature of the agents' trajectories offers realistic complexity for understanding agents actions over time instead of one-shot decisions and the systematic evaluation across 20 models provides substantial comparison points across model families and architectures. Clustering tasks that induce consistent performance across the agent group is also a clever technique for ensuring that the LLM allocator is actually faced with a dilemma, which is important to the environment.

The paper is easy to follow and the authors motivate their design decisions well. Despite the concerns I raise about the relevance of this somewhat toy setting to real-world human welfare allocation below, the task/environment captures a well-constructed and grounded decision making problem. Thus the authors findings in section 3.3 that conversational ability poorly predicts allocation competence is still interesting and informative as this suggests that the reasoning required for effective decision-making represents a distinct capability not captured by standard benchmarks. Similarly, the finding that top Arena heavily index on the initial labels of the agents' profile suggests that if allocation preferences are malleable behaviors that easily influenced by prompt engineering, we should be careful about deploying such systems in high-stakes contexts.

**Weaknesses:**

My main concern is that the features of the environment fall short of capturing the nature of high-stakes settings that affect human welfare. In particular, the lack of need-based considerations makes the study somewhat disconnected from the real welfare allocation scenarios the authors use to motivate their work.

While using knowledge/reasoning tasks might provide some insights for hiring practices where capability matters, these tasks are poor proxies for medical resource or disaster relief allocation, where the primary ethical consideration is often unrelated to capability i.e. we prioritize giving resources to those who need them most, not those who will use them most efficiently.

Additionally, the system prompt in Appendix A.6.1 shows that agents are characterized solely by their performance metrics on various benchmarks and that the allocator doesn't receive any context about more subjective factors like the agents needs, circumstances, or the moral value of ensuring they receive opportunities. In light of this, the allocator's prioritization of group productivity and heavy indexing on the performance profiles of the agents seems unsurprising (and almost preferable)-- the fundamental incentive structure seems to only reward efficiency so its not clear why the models should be fair when this would only achieve spreading tasks evenly at the cost of performance but not really benefit the agent in some way.

Overall, the current work presents an agent simulation environment that might be more useful for understanding AI-AI group dynamics than how models might impact real-world human welfare allocation.

**Questions:**

In addition to the main weaknesses mentioned above, a few smaller questions/suggestions:
- Are there any interpretable characteristics of the task flow clusters that emerged?
- Fish et al., 2025 (EconEvals: Benchmarks and Litmus Tests for LLM Agents in Unknown Environments) seems related and it would be useful to understand this paper's contributions in comparison to it.
- The output constraint experiments show models become "more utilitarian" when forced to reason briefly, but is it possible that justifying equitable distributions just requires more elaborate moral reasoning and thus more tokens, so only rather than revealing an inherent utilitarian bias in the models?
- With the addition of more diverse tasks, like writing or social reasoning, maybe the current formulation of this study be used to understand how an allocator assigns tasks to agents with different strengths and capabilities (e.g. qwen for math vs llama for writing or language-related tasks) to ensure highest group benefit.

---

### Note · Authors · 2025-12-30

I have read and agree with the venue's withdrawal policy on behalf of myself and my co-authors.